# Predicting the Year of Total Knee Replacement: A Transformer-Based Multimodal Approach

**Ozkan Cigdem**[1]                                        OZKAN.CIGDEM@NYULANGONE.ORG
[1] *Department of Radiology, New York University Langone Health, New York, USA*

**Refik Soyak**[2]                                              REFIK.SOYAK@FAU.DE
[2] *Artifical Intelligence, Friedrich-Alexander-Universität Erlangen-Nürnberg, Erlangen, Germany*

**Kyunghyun Cho**[3]                                         KYUNGHYUN.CHO@NYU.EDU
[3] *Center of Data Science, New York University, New York, USA*

**Cem M. Deniz**[1]                                      CEM.DENIZ@NYULANGONE.ORG

**Editors:** Accepted for publication at MIDL 2025

## Abstract

Accurate prediction of the year of total knee replacement (TKR) is challenging due to the complex interplay of factors influencing the surgical decision. Current deep learning models often rely on single-modality data, limiting their predictive power. Multimodal approaches integrating imaging and patient data offer the potential to improve predictions and support clinical decisions. This study presents an end-to-end trained, transformer-based multimodal model that integrates MR imaging with tabular data, including clinical variables and image readings, to predict the year of TKR for each subject. Our model leverages cross-modal attention to fuse features from an image encoder with a self-supervised pretrained tabular encoder, achieving the highest accuracy of 63.4% among tested models. We evaluated its performance against three unimodal models and four multimodal fusion strategies, including simple concatenation, DAFT, and multimodal interaction. The results demonstrate that our model's cross-modal interaction approach with pretrained TabNet not only outperformed all unimodal models but also showed improvements over other multimodal fusion techniques, highlighting the effectiveness of cross-modal attention fusion for integrating complex data modalities in TKR year prediction tasks. Source code is available at https://github.com/denizlab/2025_MIDL_time2TKR.

**Keywords:** Multimodal Learning, Year of TKR Prediction, Deep Learning, Knee Osteoarthritis

## 1. Introduction

Osteoarthritis (OA), a prevalent joint disorder, often leads to physical disability and affects global health (Kellgren and Lawrence, 1957). Knee osteoarthritis (KOA), the most common form, impacts millions worldwide, causing pain and mobility issues (Kellgren and Lawrence, 1957). It affects about 10% of men and 13% of women over 60. While there is no cure for reversing the course of KOA, total knee replacement (TKR) surgery becomes necessary in the advanced stages. Estimating the year of TKR is crucial for identifying high-risk patients and informing timely treatment decisions. However, predicting the year of TKR is complex, influenced not only by disease progression but also by individual factors like patient preferences, financial constraints, comorbidities, and overall health (Cigdem and Deniz, 2023). This variability makes accurate prediction challenging, underscoring the need for advanced predictive tools.

## 2. Related Work

In recent years, deep learning (DL) models have advanced KOA severity assessment across various imaging modalities (Rajamohan et al., 2023; Hirvasniemi et al., 2023; Tolpadi et al., 2020; Panfilov et al., 2022; Mahmoud et al., 2023). MR imaging, for example, effectively detects key structural features of knee degeneration like cartilage defects, osteophytes, joint effusion, and bone marrow edema (Rajamohan et al., 2023; Cigdem and Deniz, 2023). Most studies in the literature focus on predicting OA progression (Rajamohan et al., 2023; Panfilov et al., 2022; Leung et al., 2020; Panfilov et al., 2023), which is typically formulated as a binary classification task. Others aim to estimate OA severity (Tolpadi et al., 2020; Felfeliyan et al., 2024), KL grades (Leung et al., 2020), or symptomatic radiographic KOA (Hirvasniemi et al., 2023). These models primarily rely on imaging alone or incorporate a limited set of clinical variables, such as age, sex, BMI, and pain scores. However, only a few studies have investigated predicting the specific year of TKR, making this a relatively unexplored yet clinically important area (Mahmoud et al., 2023; Jamshidi et al., 2021; Heisinger et al., 2020; Liu et al., 2022). Previous studies using the Osteoarthritis Initiative (OAI) dataset—a 10-year observational study—have employed survival analysis methods and relied on clinical variables and image readings to estimate the year of TKR (Mahmoud et al., 2023; Jamshidi et al., 2021; Heisinger et al., 2020). However, these studies have typically focused on timeframes of no more than five years. While many approaches have demonstrated success in OA progression prediction (Rajamohan et al., 2023; Tolpadi et al., 2022; Hirvasniemi et al., 2023), existing DL models are predominantly unimodal, focusing only on imaging data without incorporating tabular data such as patient demographics, clinical assessments, or image readings (Rajamohan et al., 2023; Tolpadi et al., 2020; Panfilov et al., 2022). As physicians rely on both imaging and clinical data for accurate diagnosis, there is a growing need for automated multimodal AI systems that integrate medical images with clinical patient data to enhance consistency and precision in OA management strategies.

As tabular data gains prominence in multimodal learning, its integration becomes crucial for enhancing diverse applications (Kita et al., 2023; Felfeliyan et al., 2024; Du et al., 2024). In (Kita et al., 2023), TabNet (Arik and Pfister, 2019) and a DL model were combined for spinal cord tumor diagnosis through concatenated outputs. (Felfeliyan et al., 2024) employed a CLIP-style vision-language model to predict OA severity by merging knee radiographs with tabular OA scores. Meanwhile, (Du et al., 2024) introduced a transformer-based multimodal framework using 2D short-axis cardiac MR images and tabular data, leveraging self-supervised learning to manage missing data in cardiac disease classification.

Unlike previous studies that primarily focus on predicting KOA progression (Rajamohan et al., 2023; Panfilov et al., 2022; Leung et al., 2020; Panfilov et al., 2023), KL grade (Leung et al., 2020), or KOA severity (Hirvasniemi et al., 2023), this study specifically aims to predict the year of TKR. To this end, we developed an end-to-end, transformer-based multimodal model that integrates MR scans with clinical and image reading data. As a result, this study should not be directly compared to research focused on OA progression or severity prediction, as it addresses a distinct clinical outcome. Compared to our previous study (Cigdem et al., 2024a), where we performed survival analysis using a two-stage prediction model, this study employs end-to-end training by integrating MR scans with clinical and image reading data. Additionally, we implemented 5-fold cross-validation (CV) to obtain

a more reliable estimate of the model's generalization performance. We processed tabular data with TabNet, which is then combined with image features using cross-modal attention to predict the year of TKR for each subject. The model using the OAI dataset outputs a predicted year within a 0 to 9-year timeframe, representing the estimated year of TKR surgery. The 10-year timeframe was selected based on the OAI study design (Lester, 2008). Tabular data was encoded using TabNet (Arik and Pfister, 2019), a transformer-based model that employs sequential attention to identify key features, enhancing interpretability. TabNet also leverages unsupervised pre-training to predict masked features.

Our contributions are as follows: (1) Introducing an end-to-end trained multimodal approach that combines MR data with tabular data (clinical variables and image readings) to predict the year of TKR within 9 years. (2) Implementing a transformer-based multimodal architecture, utilizing unsupervised representation learning through masked self-supervised learning for tabular data. (3) Demonstrating that integrating image data with pretrained TabNet-processed tabular data through a multimodal interaction module based on cross-modal attention improves the accuracy of predicting the year of TKR surgery.

## 3. Method

A multimodal model is proposed to predict the year of TKR using both image and tabular data. Assume $(\mathbf{X}^i \in \mathbb{R}^{H \times W \times S \times 1}, \mathbf{X}^t = [x_t^1, \ldots, x_t^N] \in \mathbb{R}^N)$ be an image-tabular pair, where $N$ is the number of selected tabular variables. When $N_a$ is the number of categorical variables, $[x_t^1, \ldots, x_t^{N_a}]$, then $(N - N_a)$ is that of continuous variables, $[x_t^{N_a+1}, \ldots, x_t^N]$. The continuous variables were standardized using the z-score normalization (Du et al., 2024). As shown in Figure 1, the model includes a CNN-based image encoder $\phi_i$, a tabular encoder $\phi_t$, and a multimodal interaction module $\psi$.

### 3.1. Image Encoder, $\phi_i$

A 3D Resnet18 model (Tran et al., 2017) was used with the sagittal fat-suppressed three-dimensional dual-echo in steady state (DESS) MR images in OAI study cohort. To improve model generalizability, we applied random cropping during training and center cropping during validation. The resulting input image sizes were set to 300x300x160. Features were extracted from the output of the last pooling layer. The image encoder produced the image representation $\mathbf{I} \in \mathbb{R}^{H' \times W' \times S' \times C}$, where $C$ is its corresponding channel dimension. An ablation study for ResNet18 selection as the image encoder is provided in the *Ablation study* section of the Supplementary Document.

### 3.2. Tabular Encoder, $\phi_t$

Out of 1224 baseline clinical variables in the OAI (Lester, 2008) database, 245 were available for over 90% of the subjects. All available image assessment measurements were utilized. A least absolute shrinkage and selection operator (Lasso) method (Tibshirani, 1996) is applied to identify the most relevant features. The regularization strength ($\alpha$) is optimized using the Optuna framework (Akiba et al., 2019) to maximize explained variance while promoting sparsity. The optimal $\alpha$ is used to finalize the selection of 31 features. Let $(\mathbf{X}^t = [\mathbf{X}^c \mathbf{X}^r] = [x_t^1, \ldots, x_t^A] \in \mathbb{R}^A,$ where $\mathbf{X}^c, \mathbf{X}^r \in \mathbb{R}^A)$ be the concatenation of clinical

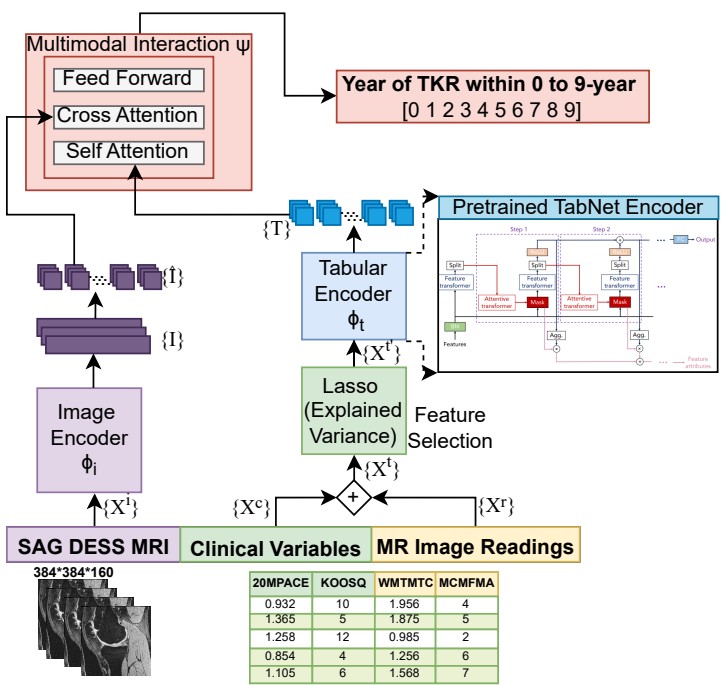

Figure 1: The architecture of the proposed model, including its image encoder, unsupervised pre-trained tabular encoder, and multimodal interaction module. DESS: sagittal fat-suppressed three-dimensional dual-echo in steady state.

variables and image readings, where A is the number of all tabular variables available in OAI dataset. After applying Lasso feature selection, the $(\mathbf{X}^{t'} = [x_t^1, \ldots, x_t^N] \in \mathbb{R}^N)$ is obtained. The masked self-supervised pretraining of the tabular encoder incorporates a task for predicting missing feature columns from the existing ones. The $N$-dimensional selected tabular features $(\mathbf{f} \in \mathbb{R}^{B \times N})$ is passed to each decision step, where $B$ is the batch size. Consider a binary mask $(\mathbf{M} \in \{0, 1\}^{B \times N})$. The encoder inputs $(1 - \mathbf{M} \cdot \hat{\mathbf{f}})$, and the decoder outputs the reconstructed features, $(\mathbf{M} \cdot \hat{\mathbf{f}})$. The prior scale term, denoting how much a particular feature has been used previously, is initialized as $\mathbf{P}[\mathbf{0}] = (1 - \mathbf{M})$ in the encoder so that the model emphasizes only the known features while the decoder's final fully-connected layer is multiplied by $(\mathbf{M})$ to output the unknown features. The reconstruction loss during the self-supervised phase is:

$$\sum_{b=1}^{B} \sum_{j=1}^{D} \left| \frac{(\hat{f}_{b,j} - f_{b,j}) \cdot S_{b,j}}{\left( \sum_{b=1}^{B} \left( f_{b,j} - \frac{1}{B} \sum_{b=1}^{B} f_{b,j} \right)^2 \right)} \right|^2. \tag{1}$$

After pretraining, TabNet leverages the learned weights and sequential attention to focus on the most relevant features at each decision step. The tabular encoder produced the tabular representation $\mathbf{T} \in \mathbb{R}^{N \times D}$, where $D$ is its corresponding channel dimensions.

### 3.3. Multimodal Interaction Module, $\psi$

A cross-attention mechanism is used to effectively capture relationships across modalities (Vaswani et al., 2017). The **I** is projected via linear layer into $\hat{\mathbf{I}} \in \mathbb{R}^{(H'W'S') \times D}$ to have the same embedding size as **T**. The interaction module is composed of $L_m$ layers, each integrating self-attention, cross-modal attention, an MLP feed-forward layer, and layer normalization. **F** captures a joint representation of an image-tabular pair. The cross-modal attention in the $l$-th layer can be formulated as described in (Vaswani et al., 2017; Du et al., 2024):

$$\text{CrossAttention}(\boldsymbol{Q}, \boldsymbol{K}, \boldsymbol{V}) = \text{softmax}\left(\frac{\boldsymbol{Q}\boldsymbol{K}^T}{\sqrt{d_k}}\right)\boldsymbol{V}, \tag{2}$$

where $\boldsymbol{Q} = \mathbf{F}^{l-1}\mathbf{W}_Q^l$, $\boldsymbol{K} = \hat{\mathbf{I}}\mathbf{W}_K^l$, $\boldsymbol{V} = \hat{\mathbf{I}}\mathbf{W}_V^l$, and $\mathbf{F}^0 = \mathbf{T}$.

## 4. Experiments

### 4.1. Study cohort

The study utilized knee data from the publicly accessible OAI database. The OAI database contains clinical variables, MRI exams, and MRI quantitative and semi-quantitative image assessment measurements for 4,796 subjects aged 45 to 79 with or at risk for KOA, evaluated at baseline and follow-ups at 12, 18, 24, 30, 36, 48, 60, 72, and 96 months. The OAI received ethical approval from the Internal Review Boards at the University of California at San Francisco. All participants provided written informed consent. The study cohort in the OAI was evaluated with longitudinal DESS MRI exams from 3.0T MRI scanner. Out of 4796 subjects in the OAI, 547 subjects underwent TKR during the 9-year follow-up period. Each subject may have undergone TKR in either one or both knees (163 with only the left knee, 168 with only the right knee, and 108 with both knees). In this study, we utilized all available data from the OAI dataset that included MR scans, image readings, and clinical data, resulting in 850 knee MRIs, as detailed in Figure 2. The baseline gender and age of study cohorts were provided in Table 1. For data augmentation, each knee was treated as an independent data point. Follow-up time point data for each patient were treated as independent, separate entries rather than part of a longitudinal study, with each follow-up time considered as year 0 for estimating the year of TKR.

### 4.2. Multimodal Model Designs

We compared the performance of our proposed model against three unimodal models, an MR image-only model, a TabNet-based tabular-only model, and a pretrained TabNet-based tabular-only model, as well as several multimodal models that integrate image and tabular data in four different ways as: *1. Basic Concatenation:* Combines image features with all tabular data via concatenation in the penultimate layer, *2. Dynamic Affine Feature Map Transform (DAFT):* Adjusts image feature maps based on tabular data, enabling images to be interpreted in the context of tabular data (Wolf et al., 2022), *3. TabNet-Processed Models:* Tabular data is processed with TabNet, then combined with image features using basic concatenation and DAFT, and *4. Self-Attention Tabular Encoder with Multimodal*

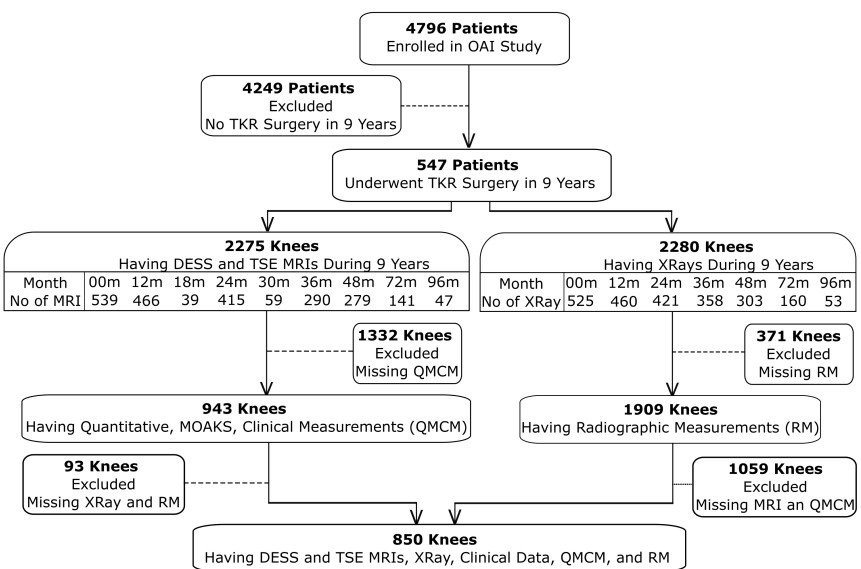

Figure 2: Flowchart for study cohort generation. After reviewing MRI data along with clinical, quantitative, and semi-quantitative image assessment measurements, 850 knee data from 547 subjects who underwent TKR within a 9-year follow-up period in the OAI database were identified. Knees: Knee images.

| Dataset | OAI |
|---|---|
| Number of Knees | 850 |
| Imaging Type | DESS |
| (Train/Validation/Test) | (604/82/164) |
| SEX | Male: 490, Female: 360 |
| AGE | Mean±SD (Range) |
| Male | 64.2±8.3 (45-83) |
| Female | 65.7±8.5 (45-82) |
| BMI | Male: 29.7±5.3, Female: 29.9±4.2 |
| RACE | Non-White: 17, White: 731, Black: 78, Asian: 24 |
| KL Grade | 0: 16, 1: 32, 2: 162, 3: 317, 4: 323 |
| PAIN (WOMAC Score) | 0: 95, 1-5: 369, 6-10: 294, 11-15: 88, 16-20: 4 |
| OARSI Grade | None: 676, Small: 43, Medium: 64, Large: 67 |
| BML Subregions | 0: 121, 1-3: 287, 4-6: 390, 7-9: 52, 10-12: 0 |

BML: bone marrow lesions, DESS: sagittal fat-suppressed three-dimensional dual-echo in steady state, OAI: osteoarthritis initiative, std: standard deviation. There are fifteen BML subregions, covering the femoral, tibial, and patellar areas. A subregion is classified as damaged if the grade is greater than 0.

Table 1: Demographic and key clinical and imaging assessment variables of subjects in the OAI study cohort.

*Interaction Fusion:* Tabular data is processed with a self-attention transformer before being combined with encoded image data using cross-modal attention (Du et al., 2024). This comprehensive comparison allowed us to identify the most effective model for predicting the

year of TKR within a 0 to 9-year timeframe by evaluating how different fusion strategies impact predictive performance.

### 4.3. Experiment Design

We used 604, 82, and 164 image-tabular data pairs for the training, validation, and test splits, respectively. The splits were made at the subject level to ensure that all follow-up data for the same subject was included within a single split. 5-fold CV was used to validate the model performance. The Kullback–Leibler divergence loss and label discretization are used for end-to-end training (Cigdem et al., 2024b). Predictions were calculated based on the area under the predicted distribution. The optimal value of the Lasso regularization parameter $\alpha$ obtained through hyperparameter tuning was 0.097. The Adam optimizer is used for all training. For self-supervised pretraining, we set the maximum epochs to 200, batch size $B = 16$, virtual batch size $B_V = 4$, masking ratio to 0.5, and a learning rate (LR) scheduler, with a starting LR of $10^{-2}$. The end-to-end model was trained with an LR of $10^{-5}$ and a weight decay of $10^{-4}$, running for 150 epochs with a batch size of 4. For TabNet, we set a categorical embedding dimension of eight, $N_d = N_a = 64$, $N_{\text{steps}} = 4$, $\gamma = 1.3$, and momentum $m_B = 0.02$. The optimal LR of 0.025, weight decay of 0.0014, nine independent layers, and seven shared layers were selected through hyperparameter tuning. Both our transformer-based tabular encoder and multimodal interaction module consist of four transformer layers, each featuring eight attention heads and a hidden dimension of 64. We used an MLP with hidden sizes of 512 for image data and 64 for tabular data, both producing outputs of size 64. The best model was selected based on the highest validation accuracy. To mitigate the risk of overfitting, we monitored validation loss and selected the model with the best validation accuracy, ensuring optimal performance. The image encoder is computationally intensive due to 3D convolutions on MRI data, while the tabular encoder remains lightweight with embeddings. The fusion module integrates both via cross-attention, reducing spatial dimensions to enhance efficiency and scalability for clinical use. The accuracy, MAE, and macro-AUC metrics were used for evaluating the models. Details of the metrics are provided in *Model prediction evaluation metrics* section of the Supplementary Document.

## 5. Results and Discussion

We evaluated the performance of two imputation methods, mean imputation for continuous variables with median imputation for categorical variables and a more advanced Random Forest-based imputation (Stekhoven and Bühlmann, 2012), alongside the impact of feature selection on model accuracy in predicting the year of TKR. Models included TabNet regressor with and without self-supervised pretraining, and the Lasso feature selection method was applied prior to encoding. As shown in Table 2, the highest accuracy of 61.0% was achieved by combining Random Forest imputation, Lasso feature selection, and the pretrained TabNet model. This configuration consistently outperformed other setups, suggesting that leveraging a feature selection and pretraining enhances model performance. Additionally, TabNet pretraining consistently improved accuracy across all settings, highlighting the benefit of self-supervised pretraining in tabular data encoding. Since we used TabNet regression to predict the time to TKR, macro-AUC could not be calculated for these models. Table 3

| Imputation Method | Feature Selection | Model | ACC (%) | MAE |
|---|---|---|---|---|
| Mean(Cont.)+Median(Cat.) | - | TabNet | 59.2 | 1.58 |
| Mean(Cont.)+Median(Cat.) | - | TabNet$_{Pretrained}$ | 60.4 | 1.55 |
| Mean(Cont.)+Median(Cat.) | Lasso | TabNet | 55.5 | 1.57 |
| Mean(Cont.)+Median(Cat.) | Lasso | TabNet$_{Pretrained}$ | 57.3 | 1.62 |
| Random Forest | - | TabNet | 54.3 | 1.65 |
| Random Forest | - | TabNet$_{Pretrained}$ | 59.8 | 1.59 |
| Random Forest | Lasso | TabNet | 57.3 | 1.74 |
| **Random Forest** | **Lasso** | **TabNet$_{Pretrained}$** | **61.0** | **1.55** |

Table 2: Performance comparison of two imputation methods, feature selection strategies, and models in predicting the year of TKR.

compares the performance of our proposed model with various multimodal and unimodal models for predicting the year of TKR within a 9-year timeframe. Among the unimodal models, the pretrained TabNet-based tabular-only model and the MR image-only model achieved similar accuracies of 61.0% and 60.7%, respectively. These results indicate that each modality alone offers competitive performance, with the pretrained TabNet particularly effective at handling tabular data independently. The proposed model combining an image encoder with a pretrained TabNet as the tabular encoder using a multimodal cross-modal attention fusion approach achieved the highest accuracy of 63.4% across all models. In comparison, basic concatenation of image data with raw tabular data and with pretrained TabNet-processed tabular data reached accuracies of 54.6% and 57.9%, respectively. DAFT showed improved performance over concatenation, achieving accuracies of 58.5% when applied to MR images with raw tabular data and 60.4% when applied to MR images with pretrained TabNet-processed data. Additionally, the multimodal model using a self-attention transformer-based tabular encoder and a cross-modal attention fusion provided an accuracy of 59.2%. As we used both knees from the same patient as independent samples, this could introduce correlation biases, potentially affecting the statistical robustness of the results. To assess this, we investigated whether training the model using only a single knee per patient would impact accuracy. This approach reduced the dataset size, leading to lower accuracy (57.7% vs. 63.4%), higher MAE (1.56 vs. 1.33), and a decrease in AUC (0.615 ± 0.040 vs. 0.665 ± 0.029). These findings reinforce the importance of leveraging both knees as independent samples to enhance model robustness and predictive accuracy.

The results underscore the advantages of our proposed model's cross-modal attention fusion approach, which outperforms other fusion methods. The highest performance achieved by our model highlights the importance of both pretraining and cross-modal attention fusion when combining tabular and image data for the year of TKR prediction over a 9-year timeframe. The proposed model is effective in capturing nuanced relationships across modalities, leading to improvements over conventional fusion techniques in multimodal learning.

TKR surgery decisions are mostly influenced by clinical symptoms, particularly pain and functional limitations, rather than radiographic OA severity alone. For instance, a patient with advanced OA (KL grade 4) may avoid surgery if they experience minimal pain, while another with mild OA (KL grade 1) may undergo TKR due to severe pain. Therefore, we use tabular data as the query vector in cross-modal attention fusion instead of imaging features. To provide a quantitative comparison, we conducted an additional experiment using image features as the query while keeping the rest of the model unchanged. This resulted in a decline in performance compared to our proposed approach, which uses tabular features as the query. Specifically, accuracy decreased from 63.4% to 57.3% MAE increased from 1.33 to 1.51 and macro-AUC dropped from $0.665 \pm 0.029$ to $0.620 \pm 0.073$.

| Data | Model | Fusion | ACC | MAE | macro-AUC $\pm$ std |
|---|---|---|---|---|---|
| Tabular-only | TabNet$_{Pretrained}$ | - | 61.0 | 1.55 | - |
| Image-only | DL | - | 60.7 | 1.46 | $0.615 \pm 0.047$ |
| Image+Tabular | DL+Raw Tabular | Concatenation | 54.6 | 1.68 | $0.539 \pm 0.058$ |
| Image+Tabular | DL+Raw Tabular | DAFT | 58.5 | 1.56 | $0.608 \pm 0.013$ |
| Image+Tabular | DL+TabNet$_{Pretrained}$ | Concatenation | 57.9 | 1.54 | $0.597 \pm 0.040$ |
| Image+Tabular | DL+TabNet$_{Pretrained}$ | DAFT | 60.4 | 1.47 | $0.644 \pm 0.024$ |
| Image+Tabular | DL+Transformer$_{Self-Attention}$ | Multimodal Interaction | 59.2 | 1.48 | $0.559 \pm 0.071$ |
| **Image+Tabular** | **DL+TabNet$_{Pretrained}$** | **Multimodal Interaction** | **63.4** | **1.33** | **$0.665 \pm 0.029$** |

Table 3: Performance comparison of the proposed model against various multimodal and unimodal models for predicting the year of TKR within a 9-year timeframe.

Our study has limitations. We included only subjects who underwent TKR within a 9-year follow-up period, requiring the pre-classification of subjects into TKR and non-TKR groups. In future work, censored data will be incorporated, and survival analysis will be conducted for the control group. Additionally, we used both knees from the same patient, which may introduce correlation; however, treating them as separate data points increases the sample size, enhances statistical power, and captures individual knee variability. While some clinical data, such as general health indicators, apply to both knees, each knee has distinct measurements and images. Since their condition and progression can vary, we treated them independently in our model. The generalizability of our model to external datasets is also limited due to differences in available clinical variables, image readings, and the absence of DESS MRI data in other cohorts, preventing direct validation. Furthermore, the OAI dataset primarily consists of older, overweight, and Caucasian subjects. As a result, the model's generalizability to populations with greater diversity in age, body mass index, race, and ethnicity requires further investigation.

## 6. Conclusion

Our study demonstrates that an end-to-end transformer-based multimodal model, integrating MR imaging and tabular data with pretrained TabNet encoder, improves the year of TKR prediction accuracy compared to unimodal and other multimodal approaches. The proposed approach can also be applied to other biomedical applications involving multimodal data integration and time-to-event analysis.

## 7. Acknowledgments

This work was supported in part by the National Institutes of Health (NIH) R01 AR074453.

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

## Supplementary Document

## S.1. Model Training

An AI model combining Resnet18 and TabNet with multi-modal cross-attention fusion was trained to predict the year of TKR surgery within a 9-year timeframe. We used data split as: 70% for training, 10% for validation, and 20% for testing. Horizontal flipping and random crop were used for data augmentation. To improve model generalizability, random cropping of input image size to 300x300x160 was implemented for DESS MR scans. Adam optimizer was used with a learning rate and a weight decay of 10-4. The model with the best validation accuracy was selected as the best model. The second last layer of Resnet18 DL model, the output of global max pooling layer before fully connected one provided 512 features for each image modality.

## S.2. Model prediction evaluation metrics

Accuracy and macro-AUC were used as estimation evaluation metrics. Accuracy was calculated as:

$$ACC = 100 \times \frac{N_{\text{correct}}}{N_{\text{total}}} \tag{3}$$

where:

- ACC: the accuracy of the TKR time prediction model

- $N_{\text{correct}}$: the number of patients whose predicted TKR time falls within $\pm 1$ year of the actual TKR time ($|y - \hat{y}| \leq 1$),

- $N_{\text{total}}$: the total number of patients in the study.

We compute the macro-AUC for a 10-class classification task, where each class represents one year to TKR (0–9 years). Since our model originally predicts 30 bins, each corresponding to 4-month intervals, we aggregate every 3 consecutive bins to obtain probabilities for 10 yearly bins before computing the macro-AUC using a One-vs-Rest (OvR) strategy. The output of our model $M \in \mathbb{R}^{B \times 30}$, where $B$ is the batch size and 30 bins correspond to 4-month intervals. Since the model outputs log-probabilities, we apply the softmax function to obtain probabilities, $P = \exp(M)$, where $P$ represents the probability distribution across 30 bins. To convert 30 bins (4-months each) into 10 bins (1-year each), we sum every 3 consecutive bins:

$$P_j^{(\text{year})} = \sum_{k=1}^{3} P_{(3j+k)} \tag{4}$$

for $j = 0, 1, ..., 9$. This gives us a new probability matrix, $P^{(\text{year})} \in \mathbb{R}^{B \times 10}$ where each column represents a 1-year probability. Let the true labels be $y$, where each ground truth $y_i$ (for the $i^{th}$ sample) represents the true time to TKR in years. The labels are discrete values, $y \in \{0, 1, \ldots, 9\}$ where each class corresponds to a yearly bin. The macro-AUC is computed using a One-vs-Rest (OvR) strategy, which involves computing AUC for each

class $k$ (treating it as a binary classification problem: Class $k$ vs. all others) and averaging the AUC scores across all 10 classes. The macro-AUC is given by:

$$\text{Macro-AUC} = \frac{1}{10} \sum_{k=0}^{9} \text{AUC}(P_k^{(\text{year})}, y_k) \tag{5}$$

where:

- $P_k^{(\text{year})}$ represents the predicted probability of class $k$,

- $y_k$ is the true label transformed into a binary format for the One-vs-Rest approach,

- AUC is the area under the receiver operating characteristic (ROC) curve.

## S.3. Ablation study

To justify image encoder choice in our end-to-end trained multi-modal model, we evaluated ResNet18, ResNet34, ResNet50, and Med3D using MRI-only data. ResNet18 provided the best prediction accuracy for our DESS MRI data from the OAI dataset, as provided in Table 4.

| Model | ACC (%) |
|---|---|
| **ResNet18** | **57.9** |
| ResNet34 | 53.1 |
| ResNet50 | 53.3 |
| Med3D | 55.8 |

Table 4: Performance comparison of AI models in predicting the year of TKR.

We compared the performance of our end-to-end trained model with commonly used traditional machine learning (ML) models for TKR prediction. Specifically, we extracted features from the image encoder and concatenated them with the selected tabular data, then evaluated the performance of a random forest (RF) model, XGBoost, and a multi-layer perceptron (MLP) using the combined dataset. Table 5 demonstrate that the end-to-end trained model outperformed these traditional ML models, highlighting the advantage of joint feature extraction and optimization in a unified framework.

| Model | ACC (%) | MAE |
|---|---|---|
| RF | 59.0 | 1.56 |
| XGBoost | 52.9 | 1.69 |
| MLP | 52.2 | 1.83 |
| **Our Model** | **63.4** | **1.33** |

Table 5: Performance comparison of ML models and our proposed end-to-end trained multimodal model in predicting the year of TKR.

