# OpenReview forum: "Predicting the Year of Total Knee Replacement: A Transformer-Based Multimodal Approach"
_MIDL.io/2025/Conference — MIDL 2025 Poster_

### Official Review · Reviewer_h3B7 · 2025-02-08

**Confidence:** 3
**Preliminary Rating:** 2
**Recommendation:** Poster
**Final Rating:** 3

**Summary:**

This paper presents a transformer-based multimodal approach to predict the year of total knee replacement. The authors use MRI and tabular data as input to the model. The model achieves an accuracy of 63.4% accuracy on the task. The authors provided pretty thorough ablations on the different components of the model, e.g., feature selection, fusion models, pretraining of TabNet.

**Strengths:**

- Predicting total knee replacement is a well-motivated clinically important task.
- Appreciate the detailed flowchart on how the dataset is obtained.
- The authors did pretty thorough ablations (e.g., feature selection, pretrain TabNet, etc.) to understand the contribution of each component to the final model performance.

**Weaknesses:**

- I'm still unsure the motivation for predicting total knee replacement surgery. Would knowledge of the date be useful for healthcare providers?
- The investigation on feature selection, data fusion etc. are interesting but brings pretty marginal gains. This paper is also not the first to proposed cross-attention fusion.
- Related to the previous point, it would be super valuable to understand the relative contribution of the imaging/tabular modality to the final prediction. For example, we see that unimodal models on MRI and on tabular data gives roughly similar performance (using MRI only is slightly better). Does this mean that clinical variables do not bring any extra useful information or is it because the existing MR reading protocole does not capture all information necessary for prediction? What kind of information do clinicians actually use (imaging vs. tabular) if presented with the same task? Further validation/discussion on these questions would be very valuable.

**Detailed Comments:**

- Why use tabular features as the query vector as opposed to using imaging features? Some qualitative argument & quantitative analysis would be helpful.
- The introduction is a bit too long with too many details not essential to getting the main motivation/idea out. It would be helpful to move a subset of the text to Related Works, Experiments. For example, readers don't need to know the experimental details of the baselines that you compared in the paper in the introduction!

**Justification Of The Final Rating:**

The authors did not meaningfully address my question regarding relative contribution of information from different modalities. However, the author's response addressed my concern wrt motivation of TKR surgery & importance of marginal gain in this case. Therefore, I'm increasing the rating to 3 Borderline.

**Justification Of The Preliminary Rating:**

The paper addresses a clinically relevant task but lacks a strong justification for why predicting the exact year of TKR is useful in practice. While the experiments are thorough, the performance gains from feature selection, fusion strategies, and TabNet pretraining are incremental. The unimodal MRI model performs nearly as well as the multimodal approach, raising concerns about the added value of using multimodal data. Further analysis would be very helpful.

**Questions To Address In The Rebuttal:**

I'm willing to raise the rating if the authors can meaningfully address weaknesses.

**Special Issue:**

No

---

> ### Author Response · Authors · 2025-03-08
>
> Predicting the time for total knee replacement (TKR) surgery is of significant clinical value, both for identifying patients at increased risk of rapid osteoarthritis (OA) progression and for providing critical prognostic insights to patients and healthcare providers. Knowing when a patient is likely to require TKR allows for early intervention strategies, such as personalized rehabilitation, lifestyle modifications, or pharmacologic treatments, which may help delay surgery or improve post-operative outcomes. Additionally, accurate TKR timing supports shared decision-making, helping patients and clinicians plan for surgery before health declines or complications arise. This ensures optimal surgical outcomes while preventing unnecessary delays or premature interventions.
>
> Undergoing TKR surgery is mostly driven by clinical symptoms, particularly pain, rather than radiographic OA severity alone. For example, a patient with KL grade 4 OA may avoid surgery if they experience minimal pain, whereas another with KL grade 1 may undergo TKR due to severe pain. Since clinical outcomes, such as pain and functional limitations, play a more decisive role in TKR decisions than imaging-based OA grading, we use tabular features as the query vector instead of imaging features.  To provide a quantitative comparison, we conducted an additional experiment using image features as the query while keeping the rest of the model unchanged. This resulted in a decline in performance compared to our proposed approach, which uses tabular features as the query. Specifically, accuracy decreased from 63.4% to 57.3% MAE increased from 1.33 to 1.51 and macro-AUC dropped from 0.665 ± 0.029 to 0.620 ± 0.073. We will incorporate these findings into the revised manuscript for further clarity.
>
> We focus on optimizing feature selection and multimodal fusion for predicting the year of TKR. Even marginal performance gains can lead to meaningful clinical improvements, helping us better identify high-risk patients. Additionally, we systematically evaluate fusion strategies, providing valuable insights into how tabular and imaging data can be effectively combined for improved prognostic modeling.
>
> Our results demonstrate that integrating diverse features from multimodal data improves the accuracy of predicting the year of TKR. The fusion of imaging, clinical variables, and image-derived readings mitigates the limitations of each modality when used alone, enabling a more comprehensive and complementary analysis. Since clinicians rely on both imaging and clinical assessments to evaluate OA, our approach aligns with real-world decision-making. However, as clinical evaluations may vary based on experience, developing an AI model provides a standardized and supportive tool to enhance decision-making consistency and accuracy.
>
> For the camera-ready version, we will revise the Introduction to make it more concise and focused on the main motivation and key contributions. As suggested, we plan to move non-essential details, such as experimental descriptions of baselines, to the Related Works and Experiments sections for better readability and organization.

---

> > ### Author Response · Authors · 2025-03-14
> > **Clarifying the Relative Contribution of Clinical and Imaging Modalities in TKR Prediction**
> >
> > Our results indicate that tabular-only models (61.0% accuracy) slightly outperform image-only models (60.7%), likely because tabular data includes both clinical variables and structured image-derived features (quantitative and semi-quantitative image readings). To better understand the impact of using only clinical or imaging data on predicting the year of TKR, we conducted additional experiments:
> >
> >    •	Clinical-only model (excluding image-derived features) achieved 59.8% accuracy.
> >
> >    •	Image-derived features only (excluding clinical variables) resulted in 57.3% accuracy.
> >
> >    •	Image-only model (raw MRI scans processed by a deep learning model) yielded 60.7% accuracy, slightly outperforming both the clinical-only and image-derived feature models.
> >
> >    •	Multimodal fusion (clinical + image-derived features + raw MRI) improved accuracy to 63.4%.
> >
> > These results indicate that clinical variables alone provide meaningful predictive value, as their accuracy (59.8%) exceeds that of image-derived features alone (57.3%). This suggests that clinical data captures additional risk factors, such as longitudinal disease progression, that are not fully represented in imaging. However, the fact that the image-only model (60.7%) outperforms both clinical-only and image-derived tabular features alone suggests that raw MRI scans contain distinct prognostic information that structured image readings fail to capture.
> >
> > The comparable performance of image-only and tabular-only models suggests that current MRI reading protocols may not fully extract all relevant prognostic biomarkers for predicting TKR within 10 years. While deep learning models can leverage latent imaging features, improvements in MRI acquisition and interpretation could enhance predictive power. This could further strengthen the discriminative capability of image-derived features and improve multimodal fusion approaches.
> >
> > Since clinicians rely on both imaging and clinical evaluations when assessing TKR risk, our findings reinforce the necessity of multimodal integration. To maximize predictive power, we combined imaging, clinical data, and image-derived features using a multimodal fusion approach, boosting accuracy to 63.4%. This demonstrates that different modalities provide complementary information, improving model performance when integrated. Imaging offers detailed structural insights, while clinical data captures pain severity, functional decline, and risk factors that might not be visible in scans. Given that clinical assessments can be subjective, AI-driven multimodal fusion provides a standardized and objective tool to enhance decision-making consistency and accuracy.

---

### Official Review · Reviewer_DNc9 · 2025-02-21

**Confidence:** 4
**Preliminary Rating:** 4
**Recommendation:** Poster

**Summary:**

This paper presents a multimodal model that predicts the year of total knee replacement (TKR) using MRI scans and clinical data. It combines a 3D ResNet18 for images and TabNet for tabular data, using cross-modal attention for better fusion. The model outperforms unimodal approaches, achieving 63.4% accuracy. While the approach is useful, the model is not novel, lacks clear justification, and is only tested on a single dataset with limited evaluation metrics.

**Strengths:**

Integrates imaging and tabular data for better TKR prediction.
Cross-modal attention improves feature fusion over simple concatenation.
Beats unimodal models, showing multimodal learning is valuable.
Tested against multiple fusion methods, adding credibility to findings.
Uses self-supervised pretraining for tabular data, improving model robustness.

**Weaknesses:**

Lacks novelty in architecture. Uses standard transformers, ResNet18, and TabNet.
No clear justification for model choice. Why use cross-modal attention over alternatives?
Dataset details are missing. No breakdown of size, diversity, or preprocessing.
Potential overfitting. Model is trained on a limited dataset without external validation.
Evaluation metrics are weak. Only accuracy and mean absolute error (MAE) are reported.

**Detailed Comments:**

Please refer to the section below

**Justification Of The Preliminary Rating:**

The paper presents a well-executed multimodal learning approach, but lacks novelty in architecture and fails to justify key design choices. The dataset details are vague, making it hard to assess generalizability. Without external validation or additional metrics, the results feel incomplete. If these gaps are addressed, the work could be stronger

**Questions To Address In The Rebuttal:**

Why was cross-modal attention chosen over simpler fusion techniques?
What is the dataset’s composition? How diverse is it?
How does this model compare to traditional machine learning methods for TKR prediction?
Have you tested this model outside the OAI dataset?

**Special Issue:**

Yes

---

> ### Author Response · Authors · 2025-03-08
>
> We acknowledge that individual components of our model, such as ResNet18, TabNet, and transformers, are widely used architectures. However, our approach is tailored for clinical relevance by integrating MRI images, clinical variables, and image readings in an end-to-end manner, reflecting how clinicians assess knee status in real-world decision-making. This multimodal integration allows for a more comprehensive and clinically interpretable prediction. To justify our model choices, we conducted extensive comparisons. For the image encoder, we evaluated ResNet18 (ACC:57.9%), ResNet34 (ACC:53.1%), ResNet50 (ACC:53.3%), and Med3D (ACC:55.8%) using MRI-only data. ResNet18 provided the best prediction accuracy for our DESS MRI data from the OAI dataset, as will be detailed in Supplementary Document Table 1. For tabular data encoding, we compared using raw tabular features, a standard ViT, and TabNet, with TabNet yielding superior performance (Table 2). For multimodal fusion, we compared direct concatenation, DAFT, and multimodal cross-attention. As shown in Table 2, the cross-attention approach achieved the best predictive performance for estimating the year of TKR. While fusing tabular and image data is not novel, our method incorporates structured clinical data, imaging-derived features, and raw images in an end-to-end trained framework.
>
> We will add information on age, sex, BMI, race, KL grade, WOMAC pain score, OARSI grade, and BML subregions for the knees from the OAI to the camera-ready version. For preprocessing, we will include the following details in the image encoder section: “To improve model generalizability, we applied random cropping during training and center cropping during validation. The resulting input image sizes were set to 300x300x160.”
>
> We compared the performance of our end-to-end trained model with commonly used machine learning (ML) models for TKR prediction. Specifically, we extracted features from the image encoder and concatenated them with the selected tabular data, then evaluated the performance of a random forest(ACC:59.0%, MAE:1.56), XGBoost (ACC:52.9%, MAE:1.69), and an MLP (ACC:52.2%, MAE:1.83) using the combined dataset. Our results demonstrate that the end-to-end trained model outperformed these traditional ML models, highlighting the advantage of joint feature extraction and optimization in a unified framework. The details of this comparison, including the ablation study, will be provided in Supplementary Document Table 2.
>
> We appreciate the reviewer’s comment. In this study, we conducted experiments using the OAI dataset, incorporating the available DESS MRI scans, clinical variables and image reading measurements. Since our model is trained in an end-to-end manner, evaluating its generalizability on an external dataset requires the same set of clinical variables, image readings, and DESS MRI data. Unfortunately, no available external dataset fully aligns with the OAI dataset in these aspects. In a separate study utilizing a two-stage model, we assessed the performance of the ResNet18 image encoder using external datasets. Specifically, we trained the model on radiographs and MRI from the OAI dataset and tested it on the external MOST MRI and radiographs, as well as an internal radiograph dataset. However, in the current study, applying the model to an external dataset would require the same set of selected features obtained through Lasso feature selection. While the MOST dataset includes clinical variables and imaging readings, they do not fully align with those in the OAI dataset, preventing a direct evaluation of the model’s generalizability.  Once the model is fully developed, the most discriminative features can be identified and prioritized for collection in future studies, enabling external validation. The primary objective of this study is to determine the most informative features using a multimodal approach. We have added the following sentence to the Discussion section to address it as a limitation: “The generalizability of our model to external datasets is also limited due to differences in available clinical variables, image readings, and the absence of DESS MRI data in other cohorts, preventing direct validation.”
>
> To mitigate the risk of overfitting, we monitored train and validation loss and selected the model with the best validation accuracy, ensuring optimal performance. This approach confirms that the selected model did not overfit, as we assessed loss trends and selected the best-performing model before any signs of overfitting appeared.
>
> In addition to accuracy and MAE, we also incorporated macro-AUC as an evaluation metric to provide a more comprehensive assessment of model performance. This metric evaluates the model’s ability to distinguish between different classes, ensuring a more robust evaluation of predictive capability. The updated results, including macro-AUC and its mathematical definition, will be reported in the manuscript.

---

### Official Review · Reviewer_7Uvq · 2025-02-22

**Confidence:** 4
**Preliminary Rating:** 5
**Recommendation:** Oral

**Summary:**

This paper gives us an interesting transformer based multi-modal framework to predict the year of total knee replacement (TKR) by integrating magnetic resonance (MR) imaging with tabular clinical data. Here, the method utilizes a 3D ResNet-based image encoder and a self-supervised pre-trained TabNet for processing clinical variables, which are subsequently fused via a cross-modal attention mechanism.Notably, the work leverages advanced techniques such as Lasso-based feature selection and unsupervised pre-training, with key formulations including the reconstruction loss.

**Strengths:**

1. The paper presents a compelling and well-motivated approach by integrating MR imaging and clinical data through an end-to-end transformer-based model.
2. The experimental section is thorough, offering a comprehensive comparison between unimodal methods and various multimodal fusion techniques, which convincingly demonstrates the superiority of the proposed approach.
3. This rigor in experimental design and the clear presentation of quantitative improvements highlights the paper’s high scientific merit and potential impact on the medical imaging and machine learning communities.

**Weaknesses:**

Despite its strengths, the paper has certain limitations that need addressing.

1. Treating both knees from the same patient as independent samples could introduce correlation biases, potentially affecting the statistical robustness of the results.
2. The absence of external validation on a dataset with greater demographic diversity leaves open questions about the model’s applicability in varied clinical settings.
3. The multimodal fusion strategy is innovative, the paper would benefit from a deeper discussion regarding the computational complexity and potential risk of overfitting, especially in scenarios with limited data?

**Detailed Comments:**

1. The manuscript is well-written and presents a clear narrative from the introduction of the problem to the detailed description of the proposed method and its evaluation.
2. Potentially - minor improvements could be made in the discussion section to further emphasize the clinical significance of accurately predicting TKR timing.

**Justification Of The Preliminary Rating:**

1. The paper presents a robust multimodal framework that fuses imaging and clinical data to predict TKR effectively. Strong experiments and deep technical insights justify a strong accept.
2. The rigorous experimental design—including comprehensive comparisons against both unimodal and alternative multimodal fusion methods—provides strong evidence of the approach.

Overall, the paper makes a significant contribution to the fields of medical imaging and machine learning, justifying a strong accept rating.

**Questions To Address In The Rebuttal:**

1. Could the authors clarify how the assumption of independence between knees from the same patient affects the overall performance, and what methods could mitigate any potential correlation biases?
2. Would the authors consider validating the model on an external dataset to verify its generalizability across a more diverse patient population?

**Special Issue:**

No

---

> ### Author Response · Authors · 2025-03-08
>
> We agree with the reviewer that the two knees of the same patient may be correlated. However, using them as separate data points increases the sample size, which can improve the statistical power of the analysis and allows us to capture individual-level variability in the data, which can be important for understanding the factors that influence the year of TKR. Each knee has different image measurements and images, which we treated as independent in our model because the condition and progression can vary between knees. For clinical data, certain questions—such as general health indicators—are indeed common for both knees, but the knee-specific measurements and imaging were independently analyzed. We have acknowledged the use of both knees from the same patient as a limitation in the Discussion section as follows:” Additionally, we used both knees from the same patient, which may introduce correlation; however, treating them as separate data points increases the sample size, enhances statistical power, and captures individual knee variability. While some clinical data, such as general health indicators, apply to both knees, each knee has distinct measurements and images. Since their condition and progression can vary, we treated them independently in our model.”  As an ablation study, we trained the model using only a single knee from a patient, specifically the one that underwent TKR earlier. The model's performance declined compared to the version where both legs were treated as separate data points (accuracy: 57.7% vs. 63.4%, MAE: 1.56 vs. 1.33, and AUC: 0.615 ± 0.040 vs. 0.665 ± 0.029). Given the limited availability of data that includes images, clinical variables, and image readings, reducing the dataset further led to a decrease in performance.
>
> In this study, we conducted experiments using the OAI dataset, incorporating the available DESS MRI scans, clinical variables and image reading measurements. Since our model is trained in an end-to-end manner, evaluating its generalizability on an external dataset requires the same set of clinical variables, image readings, and DESS MRI data. Unfortunately, no available external dataset fully aligns with the OAI dataset in these aspects. In a separate study utilizing a two-stage model, we assessed the performance of the ResNet18 image encoder using external datasets. Specifically, we trained the model on radiographs and MRI from the OAI dataset and tested it on the external MOST MRI and radiographs, as well as an internal radiograph dataset. However, in the current study, applying the model to an external dataset would require the same set of selected features obtained through Lasso feature selection. While the MOST dataset includes clinical variables and imaging readings, they do not fully align with those in the OAI dataset, preventing a direct evaluation of the model’s generalizability. Once the model is fully developed, the most discriminative features can be identified and prioritized for collection in future studies, enabling external validation. The primary objective of this study is to determine the most informative features using a multimodal approach. We have added the following sentence to the Discussion section to address it as a limitation: “The generalizability of our model to external datasets is also limited due to differences in available clinical variables, image readings, and the absence of DESS MRI data in other cohorts, preventing direct validation.”
> To mitigate the risk of overfitting, we monitored train and validation loss and selected the model with the best validation accuracy before it started to increase, ensuring optimal performance. This approach confirms that the selected model did not overfit, as we assessed loss trends and chose the best-performing model before any signs of overfitting appeared.
>
> The computational complexity of our multimodal approach is primarily driven by the image encoder (3D ResNet18), tabular encoder (TabNet), and cross-attention fusion module. The image encoder has the highest computational cost due to multiple 3D convolutional layers processing high-dimensional MRI inputs. The tabular encoder is lightweight, relying on embedding projections and attention-based feature selection. The fusion module efficiently integrates both modalities using cross-attention, with reduced spatial dimensions from the image encoder to limit computational overhead. Overall, our design balances accuracy and efficiency, ensuring scalability for clinical applications.

---

### Author Rebuttal · Authors · 2025-03-08

**Rebuttal:**

We have uploaded the clean manuscript, tracked changes version, supplementary document, and responses to the reviewers as a zip file. The responses to the reviewers have also been provided in the comment section for each reviewer.

**Supporting Material:**

/attachment/ef44fd6fd81158928e03559485480ebbfff7a77b.zip

---

### Meta-Review · Area_Chair_3Qjx · 2025-03-16

**Recommendation:** Accept (Poster)
**Confidence:** 5

**Metareview:**

This work explores the fusion of tabular data with MR imaging information to predict the year of knee replacement. The database consists of 850 patients imaged with a 3.0T MRI scanner and includes 245 clinical variables, among which 31 were ultimately selected through an automatic optimal feature selection process. A 5-fold cross-validation was performed for validation. The results demonstrate the superiority of the proposed approach compared to mono-modal methods and state-of-the-art fusion schemes. The discussion phase has been particularly productive, leading to a consensus among reviewers on the quality of this work.

For all these reasons, I have decided to accept this article.